# Comparing the levelized cost of electric vehicle charging options in Europe

Lukas Lanz[1], Bessie Noll [1] ✉, Tobias S. Schmidt [1,2] & Bjarne Steffen [2,3] ✉

With rapidly decreasing purchase prices of electric vehicles, charging costs are becoming ever more important for the diffusion of electric vehicles as required to decarbonize transport. However, the costs of charging electric vehicles in Europe are largely unknown. Here we develop a systematic classification of charging options, gather extensive market data on equipment cost, and employ a levelized cost approach to model charging costs in 30 European countries (European Union 27, Great Britain, Norway, Switzerland) and for 13 different charging options for private passenger transport. The findings demonstrate a large variance of charging costs across countries and charging options, suggesting different policy options to reduce charging costs. A specific analysis on the impacts and relevance of publicly accessible charging station utilization is performed. The results reveal charging costs at these stations to be competitive with fuel costs at typical utilization rates exhibited already today.

To mitigate climate change, the Paris Agreement of 2015 mandates a rapid and deep decarbonization of all economic sectors, including transport[1,2]. In the European Union (EU), transport was responsible for nearly a quarter (24.6%) of total greenhouse gas emissions in 2018, with road vehicles accounting for almost 75% of this[3,4]. To decarbonize this sector, fast adoption of alternative fuel vehicles is crucial[5,6].

Previous studies have acknowledged that the true cost of charging electric vehicles (EV) goes beyond uniform electricity price assumptions and must include additional factors such as charging infrastructure cost, infrastructure utilization rates, and a more accurate representation of electricity prices[7–10]. Thus far, these factors have been considered predominantly in specific contexts, such as comparison of charging cost to charging site, station technology, or drivetrain[11–13], and assessed for specific geographical locations[14–18] with limited potential for generalization of both charging option and region[19]. A recent study considers a wide range of charging options and electricity tariffs, but only for the United States[20]. For Europe with its ambitious decarbonization agenda, the cost of EV charging has not yet been modelled consistently or compared systematically across different charging options and countries, despite considerable heterogeneity of levelized cost components. The range of potential charging costs in the EU remains elusive, which is an issue particularly given the

ever more ambitious road transport decarbonization targets within the EU's 2030 emission reduction target[21,22].

European policymakers are steadily embracing EVs as a major alternative fuel vehicle option, amongst others through the European battery alliance[23–25]. Rapidly decreasing battery costs[26,27] combined with a breadth of public policy support for EVs[28,29] has dramatically reduced the initial purchase cost—a major barrier to vehicle adoption in the technology's beginnings[30]. As such, the cost of using EVs will likely gain importance, especially as car manufacturing companies continue to ramp up production, advancing the EV market from niche to mass. While operation and maintenance (O&M) costs for EVs are markedly lower than that of gasoline or diesel cars[31–34], fuel costs play an important role in the cost of using EVs, and by extension in their competitiveness with conventional internal combustion engine vehicles (ICEV)[35].

However, comparing fuel costs of ICEVs and EVs is not trivial. While gasoline and diesel costs to consumers are quite transparent (i.e. the pump price at gas stations), EV charging costs to consumers are not as straightforward, as they depend on a variety of factors including charging location, charging speed, time of charging, or even other pricing mechanisms such as charging subscription packages[7–9,36]. It is therefore not always immediately clear what charging cost an EV owner should expect when pulling up to a charging station. For transport modelers

[1]Energy and Technology Policy Group, Swiss Federal Institute of Technology, ETH Zurich, Clausiusstrasse 37, 8092 Zurich, Switzerland. [2]Institute of Science, Technology and Policy, Swiss Federal Institute of Technology, ETH Zurich, 8092 Zurich, Switzerland. [3]Climate Finance and Policy Group, Swiss Federal Institute of Technology, ETH Zurich, Clausiusstrasse 37, 8092 Zurich, Switzerland. ✉e-mail: bessie.noll@gess.ethz.ch; bjarne.steffen@gess.ethz.ch

and researchers, considering these wide-ranging differences in charging costs is key for accurate EV adoption projections. For policy-makers seeking to promote low-carbon mobility, understanding EV charging option nuances is important for informed policy intervention. Exploring charging cost in detail is therefore highly relevant.

Here we model the levelized cost of charging electric vehicles in 30 European countries and for all charging options that are relevant for EV passenger transport to address this research gap. We compile a cost component database and develop a systematic modeling framework to estimate country-specific levelized cost of charging (LCOC) of different power levels and charging sites. Importantly, detailed charging cost components are disaggregated and estimated to better understand potential cost differences and derive actionable policy implications. Taking into account cost of capital and sales margins of commercial operators, the LCOC represents the long-term average cost to the consumer and is thus compared to average costs of conventional fuels. The results underline high cost variance between modelled countries and charging options. We discuss and derive insights for transport modelers, EV users as well as policy makers for how to appropriately address EV charging prices.

## Results

Geographically, we analyze the cost of charging EVs in all 27 Member States of the EU and also include the United Kingdom, Switzerland and Norway–three additional European countries with rapidly growing EV markets. Building on the levelized cost of charging (LCOC) formula proposed by Borlaug et al.[20] we expand the approach drawing on levelized cost methods in the fields of electricity generation. The resulting measure allows for a consistent application to any charging option. We calculate the LCOC per charging plug as

$$\text{LCOC} = \left( \frac{C_{\text{equipment}} + C_{\text{installation}} + \sum_{t=1}^{T} \frac{C_{\text{O\&M}_t}}{(1+i)^t}}{\sum_{t=1}^{T} \frac{E_{\text{charging}_t}}{(1+i)^t}} + \frac{C_{\text{electricity}}}{\eta} \right) \cdot (1 + C_{\text{transaction}}) \quad (1)$$

where $C_{\text{equipment}}$ is the cost of the charging equipment hardware (€ plug$^{-1}$), $C_{\text{installation}}$ is the cost of installing the charging equipment, including all project costs except the equipment hardware (€ plug$^{-1}$), $C_{\text{O\&M}_t}$ is the cost of operation and maintenance of the charging infrastructure in year $t$ of the project's lifetime (€ year$^{-1}$), $E_{\text{charging}_t}$ is the yearly amount of energy that is charged at the plug in year $t$ of the project's lifetime (kWh year$^{-1}$), $C_{\text{electricity}}$ is the electricity cost (€ kWh$^{-1}$), $\eta$ is the charging station efficiency (%), $C_{\text{transaction}}$ is the transaction cost for settling the payment for the charging energy where applicable (%), $T$ is the project lifetime over which the LCOC is calculated (years), and $i$ denotes the interest rate, used to discount future costs and energy to a net present value (reflecting the financing cost of charging station operators where applicable) (%). By considering charging station efficiency, the model takes into account charging losses in the charging equipment. The system boundary is thus defined at the charging plug to the EV.

To analyze cost heterogeneity across different charging infrastructure, we differentiate a range of power levels and charging sites. Reflecting distinct types of charging technology and typical operating conditions in Europe, the power levels are grouped as follows:

- Low AC (<2.3 kW): socket charging without designated charging equipment (230 V, max. 10 A).
- Medium AC (3.7–7.4 kW): single-phase AC charging equipment (230 V, 16–32 A).
- High AC (11–22 kW): three-phase AC charging equipment (230 V, 3 × 16–3 × 32 A).
- DC (50 kW): DC fast charging equipment.

Note that we do not analyze charging powers above 50 kW for two reasons. First, cost data for higher power charging equipment and installation is not as readily available. In particular the grid connection cost and wholesale electricity cost for such high powered stations is less trivial. Second, higher power charging stations (>50 kW) are sparsely installed in many European countries today[37] and projections indicate that the bulk of publicly available charging stations in Europe (>80%) would be 50 kW or less through 2040[38].

In addition, the charging sites represent specific locations where charging infrastructure can be installed. We consider four different charging sites:

- Residential (grid): home charging, drawing electricity from the distribution grid.
- Residential (PV): home charging, replacing a limited share of the grid electricity during the day with electricity from a rooftop solar PV system, based on realistic hour-by-hour load profiles.
- Commercial (privately accessible): workplace or fleet charging (e.g. taxi fleet).
- Commercial (publicly accessible): publicly accessible, fee-based charging (e.g. on roads).

We define a charging option as the combination of a specific power level and charging site, as can be seen in the axes of the matrix of charging options displayed in Fig. 1. To reflect real-world charging

**Fig. 1 | Matrix of charging options with charging shares (in terms of energy) of four characteristic user profiles.** A range of power levels and charging sites are differentiated in a four by four matrix. Each combination of power level and charging site gives a charging option. We introduce four user profiles that are defined by combinations of percentage charging energy shares at different charging options: Wallbox user, Wallbox user with PV, Commercial user and Socket user. Percentage energy shares of each charging option for a given user profile are proportionally sized with bubbles. Shares within a single user profile sum up to 100% and are linked. Refer to Supplementary Table 15 for specific user profile shares. Note that the *Average user* is not shown here for better figure clarity, but can be found in Supplementary Table 15 and Supplementary Fig. 5. Reading example: The user profile *Socket user* describes an EV owner that charges 80% of the total charging energy at <2.3 kW from the residential grid and 20% at <2.3 kW from a privately accessible commercial station (e.g. at work). Source data are provided as a Source Data file.

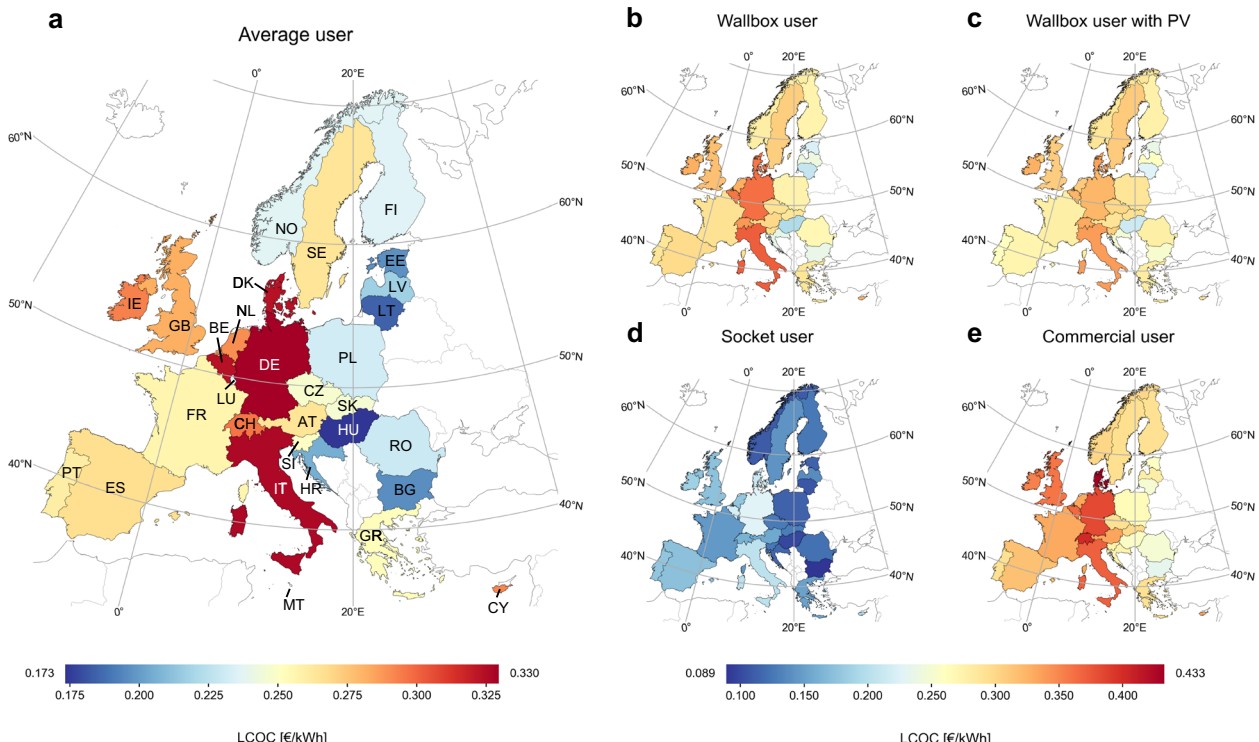

**Fig. 2 | Map of LCOC results in € per kWh of energy charged for five user profiles. a** *Average user* LCOC results. For full names of countries, please see Fig. 3. **b** *Wallbox user* LCOC results. **c** *Wallbox user with PV* LCOC results. **d** *Socket user* LCOC results. **e** *Commercial user* LCOC results. Results are displayed for all 30 modelled European countries and all units are in € kWh⁻¹. The maps are created with the Cartopy package for Python[81] and use open-source basemap data[82]. Source data are provided as a Source Data file.

behavior, we introduce user profiles, which specify the share of energy charged through different options in the total energy charged by typical EV owners. Four different characteristic user profiles are defined in Fig. 1. In addition to the four characteristic users, we define an *Average user* which represents the weighted average charging behavior of all defined users. Note that aggregating cost data from charging options to user profiles is important, as EV purchase decisions are made by users, most of which use more than one charging option. For more information on the user profiles see Supplementary Table 15.

### Variation across countries and charging options

Figure 2 depicts the LCOC for the five user profiles by country. We observe considerable variation in charging costs: The LCOC of an *Average user* ranges from 0.173 € kWh⁻¹ in Hungary to almost double that in Germany (0.330 € kWh⁻¹). Apart from Germany, Italy (0.327 € kWh⁻¹), Belgium (0.324 € kWh⁻¹), and Denmark (0.324 € kWh⁻¹) also stand out with high costs of charging. The three user profiles Wallbox user, Wallbox user with PV and Commercial user exhibit similarly large between-country variation, but all at a higher cost level, because they do not involve any low AC charging from sockets.

There are, however, important differences between these three user profiles (Fig. 2b, c, e). On a European average (weighted by population), the Wallbox user faces a cost of 0.315 € kWh⁻¹, which can be reduced to 0.300 € kWh⁻¹ if an on-site rooftop PV system substitutes grid electricity during the day. Furthermore, the profile of a *Commercial user* is quite competitive on a European average (0.333 € kWh⁻¹) and even cheaper than the *Wallbox user* profile in 6 of the 30 countries. The profile of a Socket user, which does not involve any charging infrastructure costs, is the least expensive user behavior in all countries. With a European average cost of 0.170 € kWh⁻¹, a Socket user (Fig. 2d) faces on average 43-49% lower costs than all other user profiles presented above. Note, however, that socket charging is subject to important limitations such as long charging times, necessity of socket availability over long time periods, and–in most cases–lack of smart charging options.

In Fig. 3, we dive deeper into the observed variance, disaggregating the national LCOC of user profiles into the 13 different underlying charging options. We make two observations.

First, it becomes apparent that the LCOC of different charging options varies significantly not only between but also within countries. The average difference between the highest and least cost charging option (spread) across all countries is 0.357 € kWh⁻¹. As expected, the LCOC is mainly dependent on the power level of the options–in all countries, charging at low AC stations is cheapest, DC fast charging bears the highest cost, and medium and high AC charging lies between. Interestingly, the spread of charging option costs within countries differs significantly. The minimum within country spread is observed in Romania (0.317 € kWh⁻¹), whereas the maximum spread can be found in Switzerland (0.489 € kWh⁻¹). This can largely be explained by country-specific differences between installation and grid electricity costs for the different charging locations: Generally, the within-country spread increases with lower electricity and higher installation costs.

Second, different European countries feature very distinct cost orders of charging options. On the one hand, using self-generated PV electricity for residential charging as an alternative to grid electricity can considerably reduce LCOC, especially in countries with high grid electricity tariffs such as Belgium, Denmark and Germany, or in Southern European countries that enjoy higher PV capacity factors. Italy and Cyprus combine both effects with cost reductions of 11–24% depending on the power level. The opposite is the case for the Scandinavian and especially the Baltic states (due to low PV capacity factors), as well as for some Eastern European countries (due to very low grid costs). Cost increases for residential PV versus grid charging are for instance exhibited in Norway (4 to 10% cost increase), Lithuania (12 to 23% cost increase) and Hungary (14–26% cost increase). On the

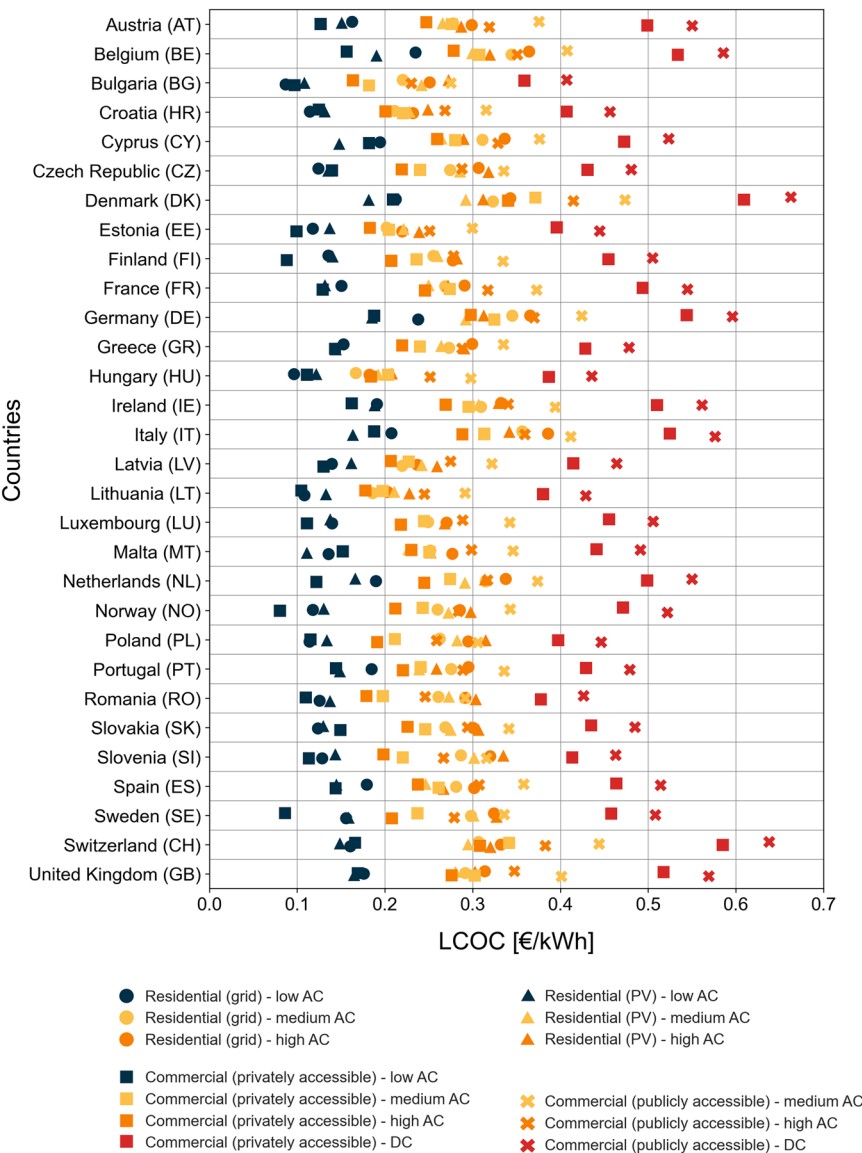

**Fig. 3 | Chart of LCOC results in € per kWh of energy charged of different charging options in all analyzed countries.** Source data are provided as a Source Data file.

hand, the attractiveness of commercial charging options compared to residential charging also depends greatly on country-specific characteristics. Charging at commercial stations is particularly opportune in most Eastern European countries, where lower yearly distances driven by private cars increases residential charging costs due to lower utilization, or in countries with significantly lower commercial electricity tariffs compared to households, for example in Belgium, the Netherlands or Sweden (see Supplementary Tables 10, 12–13). Commercial charging station costs in Eastern European countries would not necessarily be higher as a result of low yearly distances driven as stations will likely be sparsely but strategically placed based on demand to keep utilization rates high. In sum, except for low AC charging, complex interactions of all LCOC parameters create a wide range of cost distributions and rankings of charging options in the different countries.

**Drivers of variance**

To analyze how the observed LCOC diversity across European countries and charging options comes about, we disaggregate the LCOC into its components (for disaggregated results of charging options in all countries results, please refer to Supplementary Data 3). In Fig. 4, we

compare LCOC cost drivers on a European average and in a range of countries selected for their differences in relevant cost components.

On a European average, except for the user profile *Socket user*, the LCOC is composed of about half each from infrastructure (46-54 %) and electricity costs (46-54%). For the infrastructure cost category, while equipment costs are certainly relevant, installation costs generally contribute most to the total LCOC. In contrast, except for commercial public charging sites, O&M costs do not significantly affect total LCOC. For the electricity cost category, energy costs are crucial, however, at residential sites that draw grid electricity (best visible in user profiles *Wallbox user* and *Socket user*), taxes and levies on the wholesale energy price add a significant cost. The effect of involving PV is observed consistently for the selected countries—in general, on-site generation increases the energy component, but decreases costs for network and taxes.

Zooming into specific LCOC sub-categories, important cost drivers are further understood. First, the electricity category, comprised of energy and network costs as well as taxes and levies, is examined. Disregarding Switzerland, the influence of energy costs and taxes and levies is most striking—network costs are relatively constant for the four users. In general, as well for the European average, lower

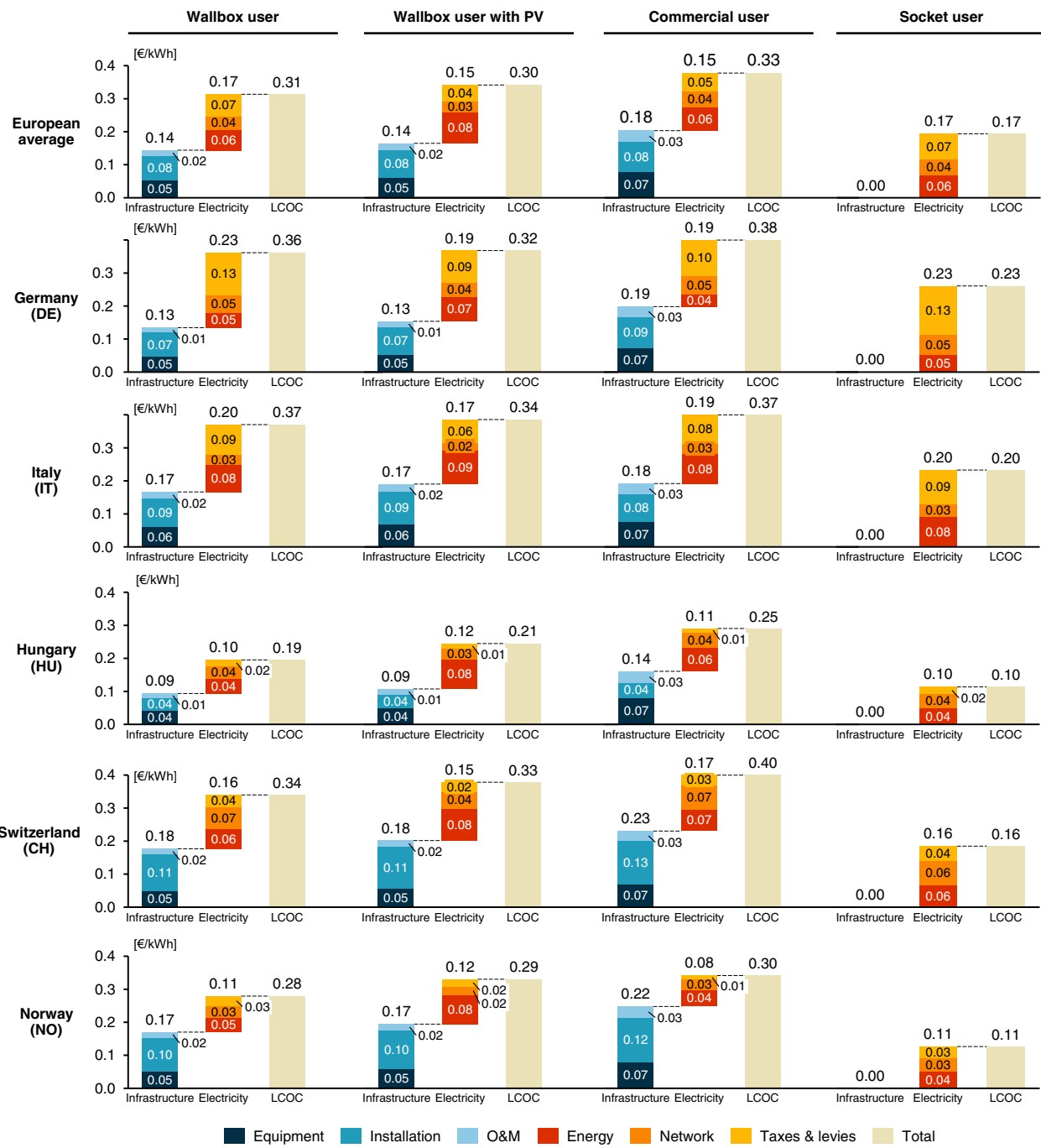

**Fig. 4 | Cost components of LCOC in different countries and user profiles in € per kWh of energy charged for the European average and selected countries.** Note that transaction costs amount to 0.004 € kWh⁻¹ at most (*Commercial user* in Denmark) and are therefore not displayed in the figure for better visibility and representation of the results. Source data are provided as a Source Data file.

electricity costs are exhibited for a *Commercial user* as compared to a *Wallbox user*. Hungary and Switzerland are two exceptions to this trend. In some countries, the profile of a *Wallbox user with PV* offers an opportunity for further cost reductions. However, using PV power at residential charging sites has both positive and negative effects. Germany shows an extreme positive effect of grid electricity avoidance primarily due to reduction of taxes and levies, which make up the largest single contribution to the total LCOC. By substituting grid electricity with PV, this cost can be significantly reduced, leading to a total decrease in LCOC of 0.040 € kWh⁻¹ (−11%), despite higher energy costs. In Hungary, on the other hand, lower costs for network and taxes when using PV power are overcompensated by the higher costs for

energy, resulting in a cost increase of 0.019 € kWh⁻¹ (+10%). Accordingly, the impact of on-site PV power generation depends not only on the levelized cost of electricity (LCOE) of PV and on the attainable charging share that can be met by PV, but also vitally depends on the cost of its substitute–grid electricity.

Second, the infrastructure category is dissected. Here the focus is mainly on equipment and installation costs, for which the within-country cost differences between the *Wallbox user* and *Wallbox user with PV* is zero (PV infrastructure costs are part of the solar PV LCOE, i.e. in the electricity category). Between the *Wallbox user* and *Commercial user*, however, infrastructure costs differ markedly: In most countries, costs increase for the *Commercial user* mainly due to

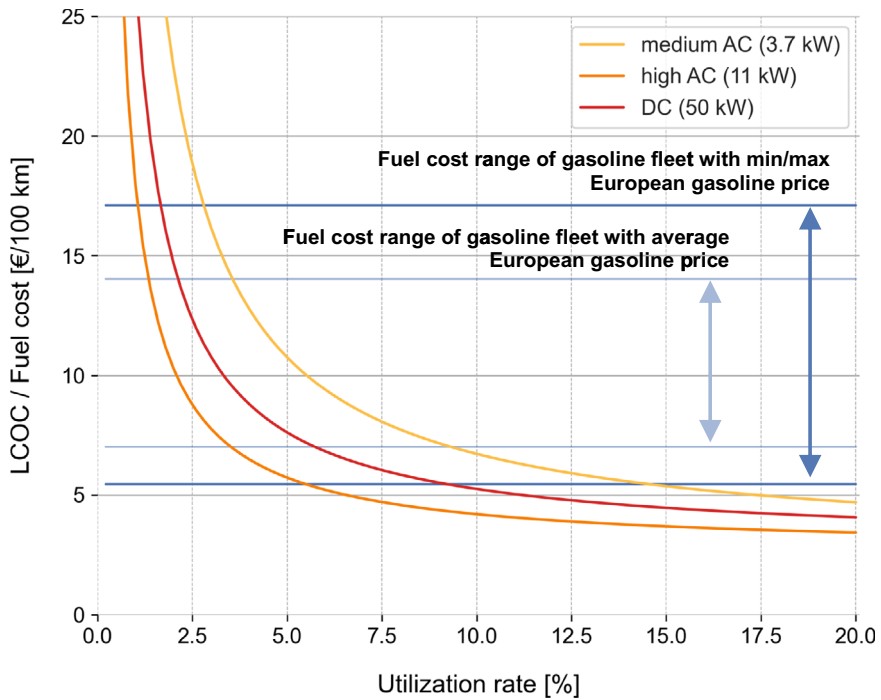

**Fig. 5 | European average LCOC in € per 100 km for three commercial, publicly accessible charging options dependent on the utilization rate of the infrastructure.** The utilization rate on the x-axis describes the hypothetical share of time that a charging station is operating at nominal power. In reality it seldom runs at fully capacity and thus the share of time the station is occupied is higher. Typical utilization rates of utilized infrastructure today are roughly 10–20% for medium AC charging, roughly 5–10% for high AC charging and roughly 1–5% for DC fast charging[20,39–47,83]. The yearly charging energies assumed in the model base case correspond to utilization rates of 10%, 5% and 4%, respectively. The blue cost bands show fuel cost ranges of today's gasoline fleet (representing around 90% of the fleet, neglecting the 5% most and 5% least efficient car models) with a 2019 average European gasoline price of 1.40 € L$^{-1}$ (inner band) and in combination with minimum (1.09 € L$^{-1}$, BG) and maximum (1.71 € L$^{-1}$, NO) 2019 yearly average gasoline prices in Europe (outer band). Source data are provided as a Source Data file.

higher equipment costs. In the case of Switzerland and Norway, which exhibit the highest labor costs among the considered countries, installation costs also play a role. This is not the case in Italy, where high equipment and relatively high installation costs for home wallboxes again makes charging away from home a similarly economic option. As compared to the German case in the previous paragraph, however, the reasoning for the Italian case is different. Due to lower annual driving distances, residential charging demand in Italy is smaller which increases levelized cost of the infrastructure. Despite similar total LCOC, the cost composition in Germany and Italy differs significantly. Likewise, Norway and Switzerland may exhibit above average infrastructure costs, but the total charging costs are below and above the European Average respectively due to differences in electricity category components. Local policy makers thus possess contrasting policy levers to reduce the LCOC, which depend foremost on key cost drivers in different countries, underpinning the importance of investigating LCOC components.

## Utilization rate

To account for input data uncertainty and to understand the most influential model factors, we conduct a sensitivity analysis of the LCOC parameters (see Supplementary Discussion 1). Based on this analysis, station utilization rate (i.e. the share of time that a charging station is operating at nominal power, see Eq. 4 in methods) emerges as a key determinant of LCOC, as it interacts with all project cost parameters and can thus lessen or exacerbate costs of equipment and installation—especially with increased capital costs at higher power levels. Apart from theoretical model sensitivities, real-world charging infrastructure features great variations concerning its utilization rate, in particular at commercial charging stations[20,39–47]. With more expensive equipment and no fixed user base, there is a higher potential risk of underutilization negatively affecting the LCOC.

In Fig. 5, the European average LCOC per 100 km of three charging options is displayed as a function of station utilization rate. The results show that the relation is highly non-linear—low utilization rates lead to very high charging costs. Additionally, a trade-off between charging speed and capital cost is observed—generally, higher power levels shift the curve downwards (lower LCOC) because the same utilization rate entails more charging energy. In contrast, the curve is shifted upwards (higher LCOC) for higher capital costs. Compared to medium AC and DC charging, high AC charging combines these two characteristics best, achieving decent charging power at relatively low cost of infrastructure. In a second step, the results are compared to fuel cost bands of conventional gasoline cars. We find that at 5–15% utilization rates, depending on gasoline prices, publicly accessible charging site costs are already cheaper than gasoline pump prices for the most efficient gasoline cars today (this of course does not take into account EV charging prices, see Discussion for clarification). Of course, when comparing the total cost of ownership (TCO) of these two vehicle types, in addition to fuel cost differences, initial vehicle purchase cost differences must also be taken into account.

## Discussion

This study contributes to the literature by comparing for the first time the levelized cost of charging electric vehicles across many charging options and countries in Europe. The results underline the high variance of costs between countries and options as well as the relative importance of different cost components.

For transport modelers, the results highlight the importance of accounting for significant charging cost heterogeneity when analyzing driving behavior, vehicle purchase decisions, or the decarbonization of passenger road transport in general. Furthermore, the results show that infrastructure cost should not be neglected. Previous studies that assume electricity prices alone as the cost to charge tend to

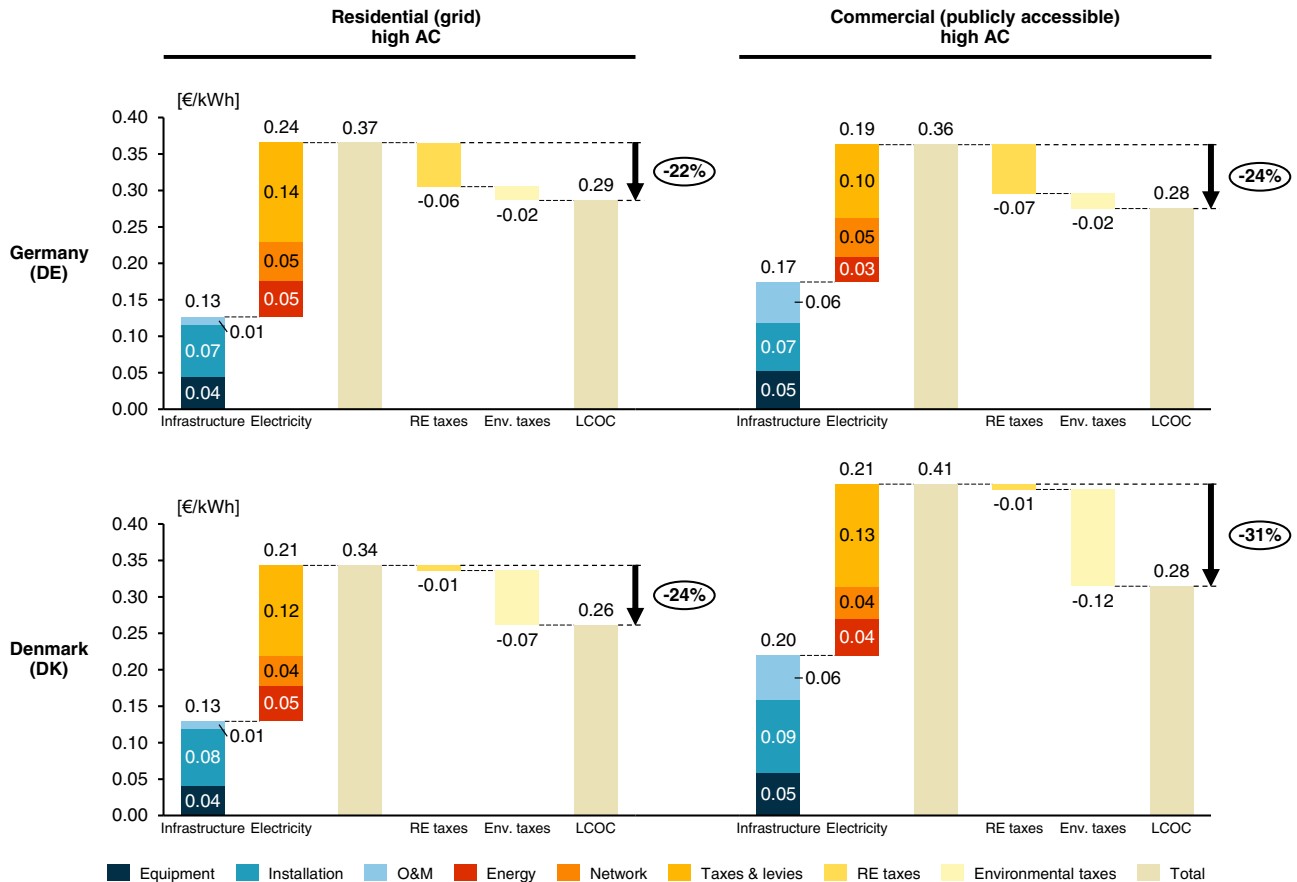

**Fig. 6 | Contributions of different cost components to LCOC of two selected charging options in Germany (DE) and Denmark (DK) in € per kWh of energy charged.** Additional segments show contributions of renewable energy (RE) and environmental (Env.) taxes and the maximum cost reduction potential of a corresponding tax exemption. Note that transaction costs are not displayed for reasons of better visibility and representation. Source data are provided as a Source Data file.

underestimate charging costs by almost a factor of two, while also neglecting the wide variance between different charging options. Without adequate reflection of the broad but dynamic charging infrastructure landscape in national contexts, EV adoption rate projections might be distorted.

For potential EV users, the vast heterogeneity of charging costs is aptly representative of the intrinsic economics of EVs versus conventional fossil fuel cars. With minor differences in the unsubsidized purchase cost of EVs across Europe, varied charging costs thus remain a decisive indicator for potential EV purchasers. However, the exhibited charging cost heterogeneity may not strictly arrive at the end-consumer. Here we modelled levelized *cost* to the end consumer only, but charging infrastructure operators might set prices differently, at least in the short term or under imperfect market environments. Such pricing schemes may depend on how charging infrastructure was historically developed, national or even sub-national-specific subsidies or price regulations, charging subscription packages, membership status, parking fees charged by the station operator, time spent charging (€ minute⁻¹) or even a flat fee for the entire charging session[48,49]. In particular, at publicly accessible commercial stations, charging prices may in fact remain rather uniform across sites, despite differences in utilization rates, in order for station operators to remain competitive. Similarly, within station operator networks uniform pricing is expected, primarily for reasons of administrative consistency. While a systemic Europe wide overview of current charging prices at publicly accessible commercial sites is not readily available, one comprehensive report for Germany in 2020 cites average charging prices from over 300 collected tariffs to be in the range of 0.30–0.35 € kWh⁻¹ [50]. In the UK in mid-2022, charging prices at publicly available

stations vary—Ionity stations charge on the high end 0.81 € kWh⁻¹ while Shell Recharge and Fastned price on the medium to low end at 0.64 € kWh⁻¹ and 0.46 € kWh⁻¹ respectively[51]. Charging prices at Swiss commercial stations are also rather varied and highly dependent on charging location (public parking lot vs. shopping mall vs. on road)—average prices in 2021 fall between 0.23-0.44 € kWh⁻¹ at medium AC powers and 0.29-0.98 € kWh⁻¹ at high DC powers[52]. We find our LCOC results compare aptly to these market values with higher general electricity prices in 2022 likely contributing to the discrepancies for high end prices in the UK and Switzerland. By and large, the diversity of charging prices across European countries, charging sites and powers as well as between station operators will influence tremendously the electric mobility transition speed in Europe.

For EV users capable of installing home-charging infrastructure, residential charging options are enticing. In certain countries, residential charging without PV power is in fact cheaper than commercial charging. For users with access to PV power generation on-site, charging costs can be further reduced, particularly in countries with high capacity factors of solar rooftop PV or high grid electricity costs. Note that this study neglects on-site storage options, which could further reduce charging costs in some cases[53]. However, these residential charging options are typically available only to house owners—EV users in city apartments mostly rely on public charging infrastructure, which often comes at higher LCOC as our analysis shows. For countries with low population shares living in owner-occupied dwellings such as Switzerland (42.5%), Germany (51.7%), Austria (55%), Denmark (61.7%) and the UK (63.4%), EV users, in particular those in lower-income groups, may rely exclusively on commercial charging and thus face much higher charging costs. This effectively exacerbates the

inequalities of EV ownership and may similarly slow the electric mobility transition in such countries.

Besides for transport modelers and potential EV users, this study bears important implications for policymakers. First, the results highlight the importance of taxes and levies within electricity tariffs: Different tax components supporting the diffusion of RE electricity generation (RE taxes) or incentivizing efficient electricity use through excise taxes (environmental taxes) can increase charging costs substantially, thus potentially inhibiting EV adoption rates. Figure 6 illustrates this concern with LCOC component breakdowns for Germany and Denmark. Notably, in Germany the RE levy is high because policymakers put great emphasis on the low-carbon transition in the electricity sector (by increasing the share of renewables that are funded by the levy), but this could hinder the low-carbon transition in the transport sector (by making the use of EVs costly vis-à-vis the use of ICE cars). Waiving these taxes or levies on electricity for EV charging would not only reduce charging costs, but also resolve inconsistencies of policy mixes targeting low-carbon transitions in the transport and electricity sectors. This would in turn increase overall policy efficiency.

Second, the results show that installation costs significantly impact the LCOC, as installation procedures remain immature and varied due to the lack of charging station installments and thus lack of installation experience in certain countries. Policy-makers may facilitate charging cost reductions by streamlining installation procedure, for example by standardizing grid-connection or by simplifying administrative processes including planning, permitting and commissioning of new infrastructure. Moreover, grid regulators should ensure that grid operators do not demand excessive grid connection fees for publicly accessible charging stations.

Third, the results suggest that policy-makers should keep an eye on electricity prices. With continuing low- or zero-emission electrification, not only in the mobility sector, electricity prices compete more and more with the price for fossil fuels. To promote the transition to EVs, policymakers should attempt to reduce electricity prices for EV charging, prevent price spikes and remove disadvantages of existing network and tax components in tariff design, for example by offering specific EV tariffs[54]. Corresponding endeavors can be observed in Germany or Slovenia[55]. Electricity price sensitivity has become an increasingly important issue in light of rising natural gas prices and the exacerbated energy crisis in Europe as a result of Russia's invasion of Ukraine. Despite the expectation of rising electricity prices, retail prices have remained rather constant since 2019, the year in which this study's price data are based. Our electricity price sensitivity analysis shows that a ±20% change in the electricity price can lead to an extreme case ±14% change in the LCOC for the Average user profile (see Supplementary Discussion 2 and Supplementary Data 4).

Finally, we underpin the recurring catch twenty-two dilemma that policy-makers in the mobility transition face: Building a dense but uncongested publicly accessible charging network ultimately drives up charging costs as a result of low station utilization rates. In turn, prioritizing well-utilized but sparsely located sites leads to a fragmented and congested, thus unattractive network. Figure 5 illustrates, however, that the goal should not be to optimize for high utilization rates (>15 %), but rather to minimize or avoid low utilization rates (<5 %) at publicly accessible charging stations whenever possible. In areas with low expected utilization rates, policy-makers can, for example, temporarily subsidize charging stations to kick-start the market, combat range anxiety and offer equitable access to public charging infrastructure while preventing excessive charging costs. For example, in Germany, tenders are organized for building a national fast charging network. Many stations are regionally distributed and different in size to involve a variety of market actors and to avoid regional monopolies. Additionally, they include both highly and poorly utilized charging locations in order to handle the mentioned dilemma of cost and coverage[56].

In sum, a wide range of points for policy intervention exist to ensure that charging costs are not a roadblock for a swift transition to electric mobility across Europe. To further develop policy advice, we outline various paths for future work. Further research is needed to expand the range of charging options, in particular concerning the rapidly emerging infrastructure of higher power levels above 50 kW. By taking a dynamic instead of static modeling approach as was presented here, cost experience curves for charging equipment, as well as for installation and O&M costs[57], could be considered and time-variant electricity prices could be taken into account. Our assumption to decrease the average electricity price by 10% to account for time-of-use tariffs may lead to an overestimation of the LCOC, particularly for residential chargers who make use of off-peak time-of-use rates, and thus constitutes a limitation of our analysis. To examine the effect of TOU electricity prices, we perform a sensitivity analysis on the 10% price decrease assumption modelling both a 20% and 30% price reduction in the Supplementary Discussion 2 (see also Supplementary Data 4). Future research may explore the potential impacts of using average prices vs. real-world tariffs on the LCOC both for residential and commercial charging locations. Third, the geographical resolution could be increased by investigating charging costs at the subnational level. Last, further research is required to incorporate the potential for different business models at residential and commercial charging stations as well as optimization of charging network utilization to reduce charging costs or allow for lower offered prices. In particular, business models that offer additional revenue streams (parking fees, retail revenues, etc.) or include vehicle-to-grid (V2G) ancillary services may further enhance the economic benefits of EV ownership. Such future research can inform decision-makers in the public and private sectors how to accelerate the low-carbon mobility transition and reach the Paris Agreement targets.

## Methods
### Cost data and assumptions
The section is structured by infrastructure and electricity cost. They form the main data input to the LCOC model and are defined for all the different charging options and countries considered in the analysis. All data sources and data processing methods are explained in detail in the Supplementary Methods. Additionally, the complete input dataset is available as a supplementary information file (see Supplementary Data 1).

**Infrastructure cost.** Adequately reflecting the different cost components of charging infrastructure is key to the levelized cost approach taken in this study. The cost of charging equipment in Europe is fragmented: A high number of original manufacturers and distribution companies offer charging infrastructure which not only differs in maximum charging power, but also in construction and technical standards. To represent the resulting cost diversity, we compile an original database of charging equipment available in Europe end of 2020, covering 232 charger models from 37 different manufacturing companies (see Supplementary Data 2). We conducted extensive desk research, requested quotes from equipment manufacturers, and complemented the price data with 12 interviews of industry experts. Drawing on in-depth research of available information and technical features of charger models, we classify the chargers into three quality standards (Home, Home/Commercial, Commercial) based on their material quality, robustness, safety standards, connectivity, communication standards and user interface. Except for the low AC power level (<2.3 kW), which represents socket charging without any designated charging hardware and thus no infrastructure cost, the equipment cost for the different charging options is determined as the average price of a slice of the database based on the respective power level and the appropriate quality standard (Home for all residential sites, Home/Commercial for privately accessible sites and both

Home/Commercial and Commercial for the publicly accessible sites). Net equipment costs are assumed to be the same across all 30 countries studied, due to the EU internal market and higher price ranges across charger models than across countries. At the two residential charging sites, country-specific VAT rates are added to all infrastructure cost components (equipment, installation, O&M). At the two commercial charging sites, VAT is ignored as commercial operators are able to reclaim it.

Second, we consider additional upfront expenses other than the charging hardware as installation cost. Mainly, these include the mechanical and structural installation of the charger (charging pole, wall mount, foundation), the establishment of an appropriate electrical distribution (grid connection, conduits, wires, trenches, safety switches, transformer for DC stations), as well as miscellaneous other expenses such as cost for planning, permitting or material transport. For high power levels, significant additional costs for bolstering the grid connection may be required. However, these additional costs vary considerably between different electricity grid area regulations and often do not reflect actual connection costs. To be as consistent as possible and to estimate comparable costs of charging, grid connection costs are therefore neglected. Within the installation cost, we differentiate costs for structural and electrical material, which are assumed to be uniform across all countries, and labor costs for construction workers and electricians, which vary by country. To determine installation costs, we collected cost information available in the literature, estimated the costs for the different charging options and then validated the cost compilation with industry experts. To account for country differences of labor costs, we estimate labor shares of the installation cost and scale the corresponding cost based on national hourly labor rates in the construction sector.

The third infrastructure component is the yearly cost for operation and maintenance (O&M) over the lifetime of a project. In general, EV charging equipment is assumed to have very low maintenance requirements. Consequently, for residential and commercial, privately accessible charging sites, we assume the O&M to be 2% of the equipment cost, covering 1-2 maintenance or repair incidents, based on expert interviews. At publicly accessible stations, we assume the O&M to be 4% of equipment cost due to higher strain of the stations. Besides the technical maintenance (O&M), for the operation of commercial, publicly accessible charging options we assume an additional 180 € per year (or 15 € per months) service cost for billing, network and load management systems, based on expert interviews.

In order to levelize the infrastructure cost, the average yearly charging energy for each charging option is required, i.e. how much a specific station is utilized throughout a year of operation. For commercial charging infrastructure we estimate average yearly charging amounts of relatively well-utilized stations. We use real-world measurements of charging behavior at privately accessible workplace or fleet charging infrastructure[39–41], publicly accessible stations[42–46] as well as specifically for DC fast charging points[20,42–44,47]. For consistency, the amount of charging energy at commercial charging sites is not differentiated between countries. For countries with emerging EV markets and low average utilizations, this analysis thus models anticipated average station usage of the near future and neglects particularly underused stations. For residential charging infrastructure, yearly charging energies of residential charging sites are estimated through a user-centered approach by projecting the yearly home charging demand of an average EV. This is approximated with average national values for yearly driven distances of private cars (km), the average real-world fuel consumption of BEVs currently sold in Europe (17.4 kWh 100 $km^{-1}$, based on the real-world fuel consumption of current BEV models sold in Europe[58,59] weighted by the models' market shares in Europe in 2019[60]) and average share of energy charged at home (75%, see also section User Profiles). Detailed information on the average yearly distances can be found in Supplementary Table 10.

We assume the lifetime of the charging infrastructure to be 15 years for all charging options based on previous LCOC studies[20], typical investment horizons and contractual arrangements, e.g. of land lease agreements. To discount yearly O&M costs and total charging energy, we assume an interest rate of 3% for residential charging sites, based on current bank interest rates for household loans and mortgages (1–5% in the Euro Area[61]) and common estimates of the social discount rate (2–3%[62]). For the commercial charging sites the cost of capital is not always consistent in literature, but for Europe, we estimate a discount rate of 7% (representing long-term averages of the cost of capital of typical companies) based on a number of studies[63–65].

**Electricity cost.** For the electricity cost, we take average grid electricity costs by country as reported by Eurostat for the residential (grid) charging site, as well as for all the commercial charging options. We complement missing countries using the same statistical approach. A list of electricity prices, detailed data sources and methods for missing data are found in Supplementary Methods 7–8. For the three charging options at the residential (grid) charging site, average electricity prices for household consumers in consumption band DD (5,000 kWh to 14,999 kWh per year) in 2019 are used, including costs for energy and supply, network costs, VAT, and all other taxes and levies. To account for the fact that most home charging takes place overnight with lower night and time-of-use tariffs, we assume the electricity cost for charging to be 10% lower than the Eurostat average. The 10% price decrease compared to the average price is determined on the assumption that night tariffs are 20% lower than the average price, day tariffs are 20% higher than the average price and that 75% of the charging happens at night (compare Supplementary Fig. 3 and 4). For the commercial options, average electricity prices for non-household consumers in consumption band IB (20,000 kWh to 499,000 kWh per year) in 2019 are used, including the same components as in the residential case except VAT.

For the residential charging site with on-site PV generation from a solar rooftop PV system in addition to the grid connection, we estimate country-specific hour-by-hour shares of the charging energy from the grid that can be replaced by the PV system during the day and build a weighted cost average. Therefore, to estimate country-specific shares of the yearly charging energy that can be supplied by the PV system, a typical hourly load curve of residential charging is modeled[66] and then overlaid with average hourly PV outputs in the different countries[67], corrected for the country's time zones and daylight saving time. The estimated shares correspond to values in similar analyses[53]. The resulting electricity cost at residential charging sites with PV is then determined for each country by averaging the levelized cost of electricity (LCOE)[68] and the residential grid tariff weighted with the determined PV share (see Supplementary Methods 4 for detailed methodology).

In order to be consistent with the infrastructure cost and to set the system boundaries of our model directly at the plug, we consider charging efficiencies of the different types of chargers. The charging efficiency is mainly dependent on the type of equipment. We assume 100% efficiency for socket charging without charging equipment (low AC) and 99.5% in models for medium and high AC charging, based on data from equipment testing[69]. For DC fast charging, we consider both the transformer and the charger itself. We assume 98% efficiency of the transformer[70] and 94% for a typical 50 kW DC fast charging station[71], resulting in an overall efficiency of 92.1%.

At publicly accessible stations, the charging service includes additional financial transaction expenses. We therefore assume an additional 2% surcharge on the total LCOC to account for a financial transaction fee incurred. For all other charging sites, the transaction cost is assumed to be 0%.

## User profiles

The different user profile charging shares are based on a broad review of measurements, models and expert studies on charging behavior and the utilization of different infrastructure[20,39,43,72–78].

The typical charging behavior of EV users in Europe is still in flux. While studies in the early EV adoption phase report a very high home charging share of 80–85%[20,39,72,75,76], mainly due to lack of publicly accessible infrastructure, this share is expected to significantly decline in the coming years, down to around 60% in the 2020 s and even well below 50% after 2030[43,77,78]. Here, we focus on an intermediate state of this transition. For the average charger we assume home-centered charging behavior with 75% of the charging occurring at home, 15% at work and 10% at publicly accessible stations. Similar trends are evident for typical EV users with regards to the charging power level: Early studies report very high shares of low AC charging (<2.3 kW), mainly because the need for higher power levels was minimal. With growing battery sizes and increasing shares of EVs, higher power levels are becoming more prevalent at many charging sites, in particular at home. We therefore assume around a third of people with access to a home charger still use low AC charging, based on empirical data[73–75] and confirmed by models of European charging infrastructure[77]. Medium and high AC charging are assumed to be equally distributed.

Based on this representation of the current average European charging behavior, four profiles of typical EV users are derived. The profile of a *Wallbox user* assumes that users own residential wallbox chargers, i.e. designated devices enabling smarter, safer and faster home charging at power levels of up to 22 kW, and that, as a consequence, low AC charging at sockets is never used, neither at home nor at work. The third profile of a *Wallbox user with PV* is identical to the *Wallbox user*, except that the residential charging electricity is assumed to be supplied, when possible, by an own rooftop solar PV system (see previous subsection). The profiles *Socket user* and *Commercial user* represent two extreme cases within the behavioral transition described above. The socket charging profile represents an early EV user that charges at home (80%) or at work (20%), does not invest in charging equipment and thus only uses low AC charging (<2.3 kW) and never uses publicly accessible charging infrastructure. The *Commercial user* represents someone without access to a home charger and thus uses commercial stations only, with a tendency to charge at higher power levels, for example an urban apartment dweller.

To compute the aggregated LCOC of these user profiles per country, the LCOC at different charging options are combined in a weighted average, see Eq. 2.

$$\text{LCOC}_{\text{country, user profile}}\left[\text{€kWh}^{-1}\right]$$
$$= \sum_{i=\text{options}} w_{\text{option }i}[\%] \cdot \text{LCOC}_{\text{country, option }i}\left[\text{€kWh}^{-1}\right] \quad (2)$$

where $\text{LCOC}_{country, option\ i}$ represents the LCOC of charging option $i$ in a specific country and $w_{\text{option }i}$ describes the charging share of option $i$ in the respective user profile. To calculate a European average LCOC, we use the population share of each country to build the weighted average of the national LCOC (see Eq. 3). The detailed list of country population used here is available in Supplementary Methods 6 (see Supplementary Table 10).

$$\text{LCOC}_{\text{European average, charging option}}\left[\text{€kWh}^{-1}\right]$$
$$= \sum_{i=\text{countries}} w_{\text{country }i}[\%] \cdot \text{LCOC}_{\text{country }i,\text{option}}\left[\text{€kWh}^{-1}\right] \quad (3)$$

where $\text{LCOC}_{country\ i, option}$ represents the LCOC of a specific charging option in country $i$ and $w_{\text{country }i}$ describes the population share of country $i$.

## Utilization rate

In Fig. 5, we show how the average European LCOC compares to the fuel cost of conventional gasoline cars, depending on the amount of yearly charging energy. To plot different power levels (and thus different amounts of yearly charging energy) in the same graph, the yearly charging energy is normalized and converted to a utilization rate. The utilization rate describes the ratio of the actual yearly energy charged to the maximum charging output that would be possible with the respective capacity, see Eq. 4.

$$\text{Utilization rate [\%]} = \frac{\text{ChargingEnergy}_t\left[\text{kWh year}^{-1}\right]}{8760\left[\text{h year}^{-1}\right] \cdot \text{nominal power[kW]}} \quad (4)$$

To compare the charging costs with fuel costs of conventional gasoline cars, both are converted to a common measure of € per 100 km, according to Eqs. 5a and 5b. Based on an analysis of the current battery electric vehicle (BEV) market, BEVs are assumed to consume 17.4 kWh 100 km$^{-1}$ on average. For an inner fuel cost range of internal combustion engine vehicles (ICEV), a European average gasoline price of 1.40 € L$^{-1}$[79] is combined with different real-world fuel consumptions representing around 90% of the range of the fleet. Fuel consumptions range from around 5 L 100 km$^{-1}$ for a lighter, efficient gasoline compact car to around 10 L 100 km$^{-1}$ for a larger and less efficient SUV[80]. To represent a more wide-ranging fuel cost range of ICEVs across Europe, the low and high fuel consumptions are combined with the minimal European gasoline price in 2019 (1.09 € L$^{-1}$, Bulgaria[79]) and the maximal European gasoline price in 2019 (1.71 € L$^{-1}$, Norway[79]), respectively.

$$\text{FuelCost}_{\text{BEV}}\left[\text{€100km}^{-1}\right]$$
$$= \text{LCOC}\left[\text{€kWh}^{-1}\right] \cdot \text{FuelConsumption}_{\text{BEV}}\left[\text{kWh }100\text{km}^{-1}\right] \quad (5a)$$

$$\text{FuelCost}_{\text{ICEV}}\left[\text{€100km}^{-1}\right]$$
$$= \text{GasolinePrice}\left[\text{€L}^{-1}\right] \cdot \text{FuelConsumption}_{\text{ICEV}}\left[\text{L }100\text{km}^{-1}\right] \quad (5b)$$

### Technical implementation

The LCOC model is implemented in Python, using an Excel spreadsheet as the main user interface for input data and modeling assumptions (see Supplementary Fig. 2 for a graphical representation of the model implementation). The complete model is provided as supplementary material in a zipped Supplementary Software file (see Supplementary Software 1). The maps representing aggregated LCOC per country (e.g. Fig. 2) are created with the Cartopy package for Python[81] and use open-source basemap data[82].

## Data availability

The authors declare that all data supporting the findings of this study are available within the paper and its supplementary information files. All input data (Supplementary Data 1), equipment cost data (Supplementary Data 2) and detailed results data (Supplementary Data 3), as well as results of the electricity price and TOU discount sensitivities (Supplementary Data 4) are provided as Supplementary Files. Source data are provided with this paper.

## Code availability

The full Python code used for this study is provided as a Supplementary Software zip file (Supplementary Software 1). The Cartopy package for Python, used to create the maps of LCOC results in Fig. 2, is publicly available for download from https://scitools.org.uk/cartopy.

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

## Acknowledgements

This project has received funding from the European Union's Horizon 2020 research and innovation programme under the Marie Sklodowska-Curie (MSC) grant agreement No 847585 (B.N.), and was sponsored by the Swiss Federal Office of Energy's "SWEET" programme and performed in the "EDGE" consortium (B.S.).

## Author contributions

This work was conceptualized by L.L, B.N., T.S.S. and B.S. The data was curated by L.L. and B.N. L.L conducted the formal analysis and visualized all results. The respective software was developed by L.L, based on previous work by B.N. L.L., B.N., T.S.S. and B.S. wrote the manuscript. T.S.S and B.S. supervised the work and acquired the necessary funding.

## Competing interests

The authors declare no competing interests.
