## [Peer Review File · Nature Communications]

REVIEWER COMMENTS

Reviewer #1 (Remarks to the Author):

KEY RESULTS:

This study extends previous literature on electric vehicle charging costs, determining the levelized cost of charging (inclusive of charging infrastructure cost, charging behaviors, and regional factors) for 30 countries in Europe. The authors find that charging costs vary widely across countries and for different charging options, more expensive in areas with high energy prices (and/or high taxes and levies) and for higher-powered and lower-utilized charging options. This study also considers the benefits (or costs) of residential photovoltaic systems for lowering (or raising) the cost of charging on a country-by-county basis, a noteworthy contribution to the existing literature.

VALIDITY:

To ensure validity and robustness of results, the authors supplement the main paper with more than 30 pages of supporting documentation. All assumptions are described in detail, referencing previous literature where data gaps exist in the public domain. Upon reviewing the materials in their entirety, I have confidence in the validity of the authors' interpretations and conclusions. Additionally, the authors present a sensitivity analysis of their model as supplemental information. This addition enhances the robustness of the study's findings, and enables valuable insights such as, the cost of residential charging is most sensitive to retail electricity prices while the cost of high power (direct current) commercial charging costs is most sensitive to station utilization (i.e., the total energy dispensed to electric vehicles).

SIGNIFICANCE:

While the purchase price for electric vehicles continues to decline, their major economic value proposition remains their lower expected operating costs (fuel and maintenance) compared to conventional petrol-powered vehicles. For researchers, businesses, and consumers to make an accurate comparison of electric vs. petrol-powered vehicles on a total cost of ownership basis, accurate geographically resolved electric vehicle charging cost estimates, inclusive of equipment purchase, installation, maintenance, and operation costs, are needed. As the authors note, while other studies have determined the levelized cost of charging in other regions of the world, this study provides the most comprehensive assessment for Europe, one of the fastest growing electric vehicle markets in the world.

DATA & METHODOLOGY:

The authors leverage previous formulas described in literature and levelized cost methods in the fields of electricity generation to calculate the levelized cost of charging for various charging technologies and at various site types. Their approach is valid and thoroughly described in an accessible manner. The use of average residential and commercial electricity prices in place of real-world tariff analysis is understandable, as this data may not be widely available across all European countries, however, it will likely lead to the overestimation of residential charging costs, since off-peak time-of-use rates typically align well with home charging, and the underestimation of commercial costs, since EV charging stations (especially early-market low utilization ones) are particularly impacted by high demand charges. The authors apply a 10% discount to the average residential electricity price to account for time-of-use tariffs, but it isn't clear how this was determined since savings during off-peak time-of-use periods are typically much greater than 10%. The authors should discuss the potential impact from using average prices vs. real-world tariffs on their cost estimates and add this as a limitation in their discussion.

ANALYTICAL APPROACH:

The analytical approach for this study is solid and results appear to be valid. I have no pressing concerns.

SUGGESTED IMPROVEMENTS:

Several discussion points and clarifications are recommended to increase the paper's impact –

1. In their list of charging technologies, the authors only consider one DC charging power level – 50 kW. This is a curious choice as the industry is already moving well past 50 kW with many units capable of up to 350 kW today. While it may be too much to ask for the authors to provide a commercial 350 kW scenario, they should at least describe why the choice was made to only model up to 50 kW and discuss how their findings might be extrapolated to higher power levels.

2. The authors should include a table as supplemental information that compares charging costs to petrol prices on a cost per 100 km basis (as computed in equations 5a and 5b) for all user profiles and regions studied.

3. On p. 7 – “Charging at commercial stations is particularly opportune in most Eastern European countries, where lower yearly distances driven by private cars increases residential charging costs due to lower utilization...” <- Wouldn't less driving also imply lower utilization and higher costs for commercial charging?

4. On p. 12 – “However, these residential charging options are typically available only to house owners–EV users in city apartments mostly rely on public charging infrastructure, which often comes at higher LCOC as our analysis shows”. <- This is a key finding that I wish the authors would have expanded upon. What percentage of the population in each of these countries are single-family homeowners vs. renters and/or apartment dwellers? The one characteristic user profile that relies exclusively on commercial charging has the highest average charging costs. They’re also likely the group with the lowest income. If low-income electric vehicle drivers are forced to pay more for charging, it will take longer to (or they may never) reap the total cost of ownership benefits of electric vehicles. This will only exacerbate the inequalities of electric vehicle ownership.

CLARITY & CONTEXT:

The authors have done a good job writing in a clear and accessible manner. Their conclusions are made in sufficient context to previous work as well as their own.

REFERENCES:

The authors have done well to summarize the previous literature in a succinct manner.

Reviewer #2 (Remarks to the Author):

The authors present a very interesting study with the aim of estimating the levelized cost of charging EVs across different European countries, considering additional aspects driving the variability of the costs. While the study is innovative, and based on a sound scientific methodology, I believe that some important aspects need to be addressed before this paper can be considered for publication.

The main aspects are the following:

The calculations are performed on 2019 electricity prices, which seem already outdated considering the current situation in Europe and the future expectations. Moreover, this value seems to be considered constant over the lifetime of the equipment (as a result from the formulation of the LCOC in equation 1). I am wondering the impact of such a choice on the significance and reliability of the results presented in this work.

I believe there is an issue of comparability when considering households and commercial charging points. While in household charging the final cost for the users should be the LCOC calculated in the paper, in commercial charging the LCOC represents the cost for the supplier, and not for the final users. Thus, these two costs are actually not directly comparable (nor can they be directly summed up for user profiles with a mixed household/commercial charging). Please address this issue.

In parallel, I believe a comparison of the LCOC calculated for the commercial charging with the average price that is charged to users could provide interesting insights to be included in the discussion.

The role of PV generation in households is not clear to me. In Figure 1 the two user profiles “Wallbox” and “Wallbox with PV” seem identical, which makes sense. At the same time, the “Wallbox with PV” will likely not use all its household-electricity from PV directly, but a significant amount of charging time will happen at night, when PV is not available (if not, I expect the shares to be different for the two user profiles). Many EU countries have different rules for self-consumption accounting, so the authors should clarify if they consider actual electricity from PV (and at what price) or an annual balance between production and consumption (without considering different hourly profiles for charging and PV generation).

Additional minor aspects to be addressed:

Line 42: it is true that fuel prices are transparent, but so are EV charging prices in most public chargers. Moreover, also traditional fuels show a significant variability (by country, location, highways, brand, etc.), just like for EV charging. This should be maybe further clarified. On the other hand, fuel costs are constantly varying, while in many cases charging costs show a certain level of stability. This aspect may be worth being mentioned as well.

Lines 49-51: It is not clear to me if the authors are discussing charging prices for the users or charging costs for the suppliers, as they are often mixing these two concepts. Please double-check these paragraphs to provide a coherent interpretation.

Equation 1: it is not clear to me why the O&M annual costs are discounted while the electricity cost is not. I assume this is coherent with other levelized cost methods in the field. Still, the authors may also want to briefly discuss the effects of this choice.

Figure 5: As stated above, LCOC is actually representing the cost for the supplier, not for the final users. Thus, it cannot be directly compared with fuel costs (that include the revenue of the suppliers). Please address this issue.

Response to Peer Review	
--

Manuscript Number	NCOMMS-21-37909
Manuscript Title	Comparing the levelized cost of electric vehicle charging options in Europe
Peer Review received	March 15, 2022

Response to Reviewer # 1

No.	Reviewer's comment	Answer
1.1	Key Results: This study extends previous literature on electric vehicle charging costs, determining the levelized cost of charging (inclusive of charging infrastructure cost, charging behaviors, and regional factors) for 30 countries in Europe. The authors find that charging costs vary widely across countries and for different charging options, more expensive in areas with high energy prices (and/or high taxes and levies) and for higher-powered and lower-utilized charging options. This study also considers the benefits (or costs) of residential photovoltaic systems for lowering (or raising) the cost of charging on a country-by-county basis, a noteworthy contribution to the existing literature.	We thank the reviewer for her/his time devoted to the review, and are happy to hear the positive overall remark. The reviewer's comments have been very helpful to further improve the quality of the paper. We revised the manuscript taking all the reviewer's comments into account. Below you will find detailed answers to each comment concerning the different sections of the manuscript.
1.2	Validity: To ensure validity and robustness of results, the authors supplement the main paper with more than 30 pages of supporting documentation. All assumptions are described in detail, referencing previous literature where data gaps exist in the public domain. Upon reviewing the materials in their entirety, I have confidence in the validity of the authors interpretations and conclusions. Additionally, the authors present a sensitivity analysis of their model as supplemental information. This addition enhances the	We are happy to hear the reviewer found the validity of the manuscript sound and that the sensitivity analyses provided in the supplemental information helped to enhance the robustness of the study.

	robustness of the study's findings, and enables valuable insights such as, the cost of residential charging is most sensitive to retail electricity prices while the cost of high power (direct current) commercial charging costs is most sensitive to station utilization (i.e., the total energy dispensed to electric vehicles).	
1.3	Significance: While the purchase price for electric vehicles continues to decline, their major economic value proposition remains their lower expected operating costs (fuel and maintenance) compared to conventional petrol-powered vehicles. For researchers, businesses, and consumers to make an accurate comparison of electric vs. petrol-powered vehicles on a total cost of ownership basis, accurate geographically resolved electric vehicle charging cost estimates, inclusive of equipment purchase, installation, maintenance, and operation costs, are needed. As the authors note, while other studies have determined the levelized cost of charging in other regions of the world, this study provides the most comprehensive assessment for Europe, one of the fastest growing electric vehicle markets in the world.	We appreciate this comment from the reviewer and are encouraged by the satisfaction with the significance as well as the contribution to literature.

1.4	Data and Methodology: The authors leverage previous formulas described in literature and levelized cost methods in the fields of electricity generation to calculate the levelized cost of charging for various charging technologies and at various site types. Their approach is valid and thoroughly described in an accessible manner.	We thank the reviewer for this comment and are happy to hear our LCOC method is sound and clearly described.
1.5	Data and Methodology: The use of average residential and commercial electricity prices in place of real-world tariff analysis is understandable, as this data may not be widely available across all European countries, however, it will likely lead to the overestimation of residential charging costs, since off-peak time-of-use rates typically align well with home charging, and the underestimation of commercial costs, since EV charging stations (especially early-market low utilization ones) are particularly impacted by high demand charges. The authors apply a 10% discount to the average residential electricity price to account for time-of-use tariffs, but it isn't clear how this was determined since savings during off-peak time-of-use periods are typically much greater than 10%. The authors should discuss the potential impact from using average prices vs. real-world tariffs on their cost estimates and add this as a limitation in their discussion.	We thank the reviewer for this insightful comment. In early discussions of how to deal with time-of-use tariffs (TOU), we indeed considered implementing these real world effects, but ultimately decided against it because – as the reviewer suggests – the available data was limited for a complete analysis of all considered European countries. The 10% price decrease assumption was determined based on the assumption that night tariffs are 20 % lower than the average price, day tariffs are 20 % higher than the average price and that 75 % of residential charging happens overnight. This assumption was made based on a few sources and as well on market research of electricity tariffs in different European countries. We have added a few more lines on our reasoning for the 10% selection in the Methods section in paragraph 1 of the “Electricity cost” section of the main manuscript (Pg. 18). Pg. 18 The 10 % price decrease compared to the average price is determined on the assumption that night tariffs are 20 % lower than the average price, day tariffs are 20 % higher than the average price and that 75 % of the charging happens at night (compare Supplementary Figure S3). Additionally, to account for the potential limitations of this 10% discount assumption, we include newly a 20% and 30% TOU tariff discount sensitivity in the Supplemental Information B.1.2, Supplementary Table S18. We’ve also included new figures for the sensitivities (Supplementary Figure S12-S14). For the “Average User”, a 20% discount assumption results in a ~5% decrease of the LCOC and a 30% discount assumption results in a ~10% decrease of the LCOC. Only for the “Socket user” profile does the sensitivity show a high end 20% decrease of the LCOC (under the 30% discount assumption). Interestingly, we observe the biggest changes between the residential (grid) and residential (PV) LCOC in situations where the grid electricity share is larger than the PV electricity share for residential overnight charging. In some country specific cases, this means the cost order of residential (grid) and residential (PV) switches, e.g. Ireland, Luxembourg (at the 20% TOU discount); Austria, Greece, Switzerland, UK (at the 30% TOU discount). Furthermore, the LCOC cost order of residential vs. commercial sites changes in many countries as well. For example, in countries where commercial (public) charging is cheaper under base TOU tariff assumptions (i.e. Cyprus, Czech Republic, Greece, Italy, Netherlands, Portugal, Slovakia, Norway), residential (grid) charging becomes cheaper under 30% TOU tariff discount assumptions. Generally, this trend indicates that home charging becomes even

		more advantageous than commercial charging under more “realistic” TOU tariff assumptions. We discuss newly the limitations of the 10% price decrease assumption as well as the 20 and 30% price decrease sensitivities in the future research part of the discussion section (Pg. 15). Pg. 15 Our assumption to decrease the average electricity price by 10% to account for time-of-use tariffs may lead to an overestimation of the LCOC, particularly for residential chargers who make use of off-peak time-of-use rates, and thus constitutes a limitation of our analysis. To examine the effect of TOU electricity prices, we perform a sensitivity analysis on the 10% price decrease assumption modelling both a 20% and 30% price reduction in the Supplemental Information B.1.2 (Table S18). Future research may explore the potential impacts of using average prices vs. real-world tariffs on the LCOC both for residential and commercial charging locations.
1.6	Analytical Approach: The analytical approach for this study is solid and results appear to be valid. I have no pressing concerns.	We are happy to hear the reviewer finds our analytical approach valid.
1.7	Suggested Improvements 1: In their list of charging technologies, the authors only consider one DC charging power level – 50 kW. This is a curious choice as the industry is already moving well past 50 kW with many units capable of up to 350 kW today. While it may be too much to ask for the authors to provide a commercial 350 kW scenario, they should at least describe why the choice was made to only model up to 50 kW and discuss how their findings might be extrapolated to higher power levels.	We thank the reviewer for this comment and would like to stress that this was indeed a big point of discussion for us in the early stages of our analysis— what range of charging station power levels should our study cover. While we agree with the reviewer that the general industry trend of EV charging station installation is moving towards higher power 150kW or even 350kW DC public stations, this is not the reality in most European countries today. We would like to highlight two reports in particular that back this statement: the first from the European Automotive Manufacturers Association (ACEA) who recently released a study finding that only 1 in 9 charging points in the EU is “fast” (defined as >2kW)¹. The second report is from Bloomberg New Energy Finance (BNEF), which projects charging infrastructure roll-out for passenger vehicles and vans in the EU to be predominantly (>80%) comprised of 7-22kW charging stations through 2040 (see figure below)². Together, these two reports emphasize the scarcity of high-powered charging stations today and likely also in the near future. As this study is foremost intended to represent a current “snapshot” of EV charging costs, we felt it outside the scope of the study to include charging powers over 50kW. Secondly, data availability for both equipment and installation costs of higher than 50kW power charging stations is limited. In particular the grid connection cost and wholesale electricity cost for such high powered stations is less trivial. The wholesale electricity cost especially would depend more on

¹ ACEA, *Making the Transition to Zero-Emission Mobility*, 2021 [<https://www.acea.auto/publication/2021-progress-report-making-the-transition-to-zero-emission-mobility/>]

² BNEF, *Electric Vehicle Outlook*, 2021 Report

		the load profile of the station thus making it difficult to justify using Eurostat values for electricity prices—additional assumptions would have to be made regarding time-of-use tariffs as noted by the reviewer in response comment No. 1.5. We include a few additional lines in the introduction section of the manuscript to highlight why higher charging station power beyond 50kW were not included in the study (Pg. 3). Pg. 3 Note that we do not analyze charging powers above 50kW for a couple of reasons. First, cost data for higher power charging equipment and installation is not as readily available. In particular the grid connection cost and wholesale electricity cost for such high powered stations is less trivial. Second, higher power charging stations (>50kW) are sparsely installed in many European countries today and projections indicate that the bulk of publically available charging stations in Europe (>80%) would be 50kW or less through 2040. Figure 324: Charging infrastructure roll-out for passenger vehicles and vans in Europe to 2040 Source: BNEF. Note: Annual public installations in 2020 may differ from those highlighted in other BNEF reports as the outlook does not include 3kW chargers or CHAdeMO fast charge connectors. Annual private installations are estimated. Figure Source: BNEF (2021)
1.8	Suggested Improvements 2: The authors should include a table as supplemental information that compares charging costs to petrol prices on a cost per 100 km basis (as computed in equations 5a and 5b) for all user profiles and regions studied.	We thank the reviewer for this suggestion. We have now included this table in the Supplemental Information B.2 (Supplemental Table S20).
1.9	Suggested Improvements 3: On p. 7 – “Charging at commercial stations is particularly opportune in most Eastern European countries, where lower yearly distances driven by private cars increases residential charging costs due to lower utilization...” <- Wouldn't less driving	This is an interesting point raised by the reviewer. While it is true that fewer annual kilometers would imply lower utilization and thus a higher LCOC at commercial stations, this does not account for the fact that commercial charging stations can be strategically placed based on demand. In Eastern European countries, this would likely be the case as charging station operators would not install stations that they expect to be used infrequently. A quarter of all EV charging points in Europe are concentrated in just four European countries (Netherlands, Germany, France, United Kingdom), while commercial charging stations in Eastern Europe are incredibly sparse³. As EV sales penetration in Easter European countries also remains low, strategic station placement becomes ever more important for countries such as Greece,

³ ACEA, *Making the Transition to Zero-Emission Mobility*, 2021 [https://www.acea.auto/publication/2021-progress-report-making-the-transition-to-zero-emission-mobility/]; ChargeMap [https://chargemap.com/map]

	also imply lower utilization and higher costs for commercial charging?	Cyprus, Malta, Bulgaria and Romania in order for station operators to maintain profitability. Commercial charging stations in Eastern Europe may therefore have relatively high utilization rates and thus lower costs, simply because fewer stations exist. For EV owners in Eastern Europe who like to charge primarily at home, the sunk investment cost of the home charging equipment together with inherently low utilization rates from a user base of only one or two vehicles leads to high charging costs. In order to avoid confusion with regard to the reviewer identified statement, in the manuscript in the “Variation across countries and charging options” subsection of the Results, we add a few additional sentences to make the above discussed points clear (Pg. 8). Pg. 8 Commercial charging station costs in Eastern European countries would not necessarily be higher as a result of low yearly distances driven as stations will likely be sparsely but strategically placed based on demand to keep utilization rates high.
1.10	Suggested Improvements 4: On p. 12 – “However, these residential charging options are typically available only to house owners—EV users in city apartments mostly rely on public charging infrastructure, which often comes at higher LCOC as our analysis shows”. <- This is a key finding that I wish the authors would have expanded upon. What percentage of the population in each of these countries are single-family homeowners vs. renters and/or apartment dwellers? The one characteristic user profile that relies exclusively on commercial charging has the highest average charging costs. They’re also likely the group with the lowest income. If low-income electric vehicle drivers are forced to pay more for charging, it will take longer to (or they may never) reap the total cost of ownership benefits of electric vehicles. This will only exacerbate the inequalities of electric vehicle ownership.	This suggested improvement from the reviewer is very much appreciated. Access to residential charging is indeed advantageous for EV owners in some European countries in terms of cost to charge and a key finding of our results. EV owners who rely exclusively on commercial charging will face much higher charging costs, which leads to higher a total cost of ownership thus exacerbating the inequalities of EV ownership as the reviewer points out. This would also speak to relatively slower EV transition speeds in the passenger sector for European countries with low population shares living in owner-occupied dwellings—particularly for low-income groups in these countries who would rely heavily on high-cost commercial charging options. To complement the point made on page 12 of the manuscript, we have collected additional country-specific auxiliary data on share of population living in owner-occupied dwellings. This data is now included newly in the Supplemental Information A.6.1, Supplementary Table S11. Further, we comment newly on this data in the discussion section following the original comment on residential charging options typically being available only to house owners (Pg. 13). Pg. 13 For countries with low population shares living in owner-occupied dwellings such as Switzerland (42.5%), Germany (51.7%), Austria (55%), Denmark (61.7%) and the UK (63.4%), EV users, in particular those in lower-income groups, may rely exclusively on commercial charging and thus face much higher charging costs. This effectively exacerbates the inequalities of EV ownership and may similarly slow the electric mobility transition in such countries.

1.11	Clarity & Context: The authors have done a good job writing in a clear and accessible manner. Their conclusions are made in sufficient context to previous work as well as their own.	We are happy the reviewer was satisfied with the writing as well the contextualization of the discussion section.
1.12	Reference: The authors have done well to summarize the previous literature in a succinct manner.	We are happy to hear the literature review was satisfactory to the reviewer.

Response to Reviewer # 2

No.	Reviewer's comment	Answer
2.1	Remarks to the Authors: The authors present a very interesting study with the aim of estimating the levelized cost of charging EVs across different European countries, considering additional aspects driving the variability of the costs. While the study is innovative, and based on a sound scientific methodology, I believe that some important aspects need to be addressed before this paper can be considered for publication.	We thank the reviewer for devoting their time to the review, and are happy to hear that she/he considers the manuscript to be interesting and useful. We appreciate the concerns raised by the reviewer and address them in detail below.
2.2	Major Aspects 1: The calculations are performed on 2019 electricity prices, which seem already outdated considering the current situation in Europe and the future expectations. Moreover, this value seems to be considered constant over the lifetime of the equipment (as a result from the formulation of the LCOC in equation 1). I am wondering the impact of such a choice on the significance and reliability of the results presented in this work.	As indicated in the response to the editor (point No. 0.1), we address the concern of variable electricity prices through a sensitivity analysis. The sensitivity analysis examines the impact of high and low electricity prices (+/- 20%) as compared to the base 2019 prices that were originally used. We reviewed the average EU electricity prices for the years 2020 and 2021 and in fact found very little variation as compared to the 2019 prices. The average EU electricity price (all taxes and levies included) in the household consumption band DD (our assumption for home charging sites) for 2019, 2020 and 2021 were 0.198, 0.196, and 0.208 EURO per kWh respectively. For the non-household consumption band IB (our assumption for commercial charging sites), the average EU electricity prices for 2019, 2020 and 2021 were 0.179, 0.183 and 0.196 EURO per kWh respectively. Despite the expectation of rising electricity prices, in particular because electricity wholesale prices are strongly affected by gas prices which have soared most recently as a result of the war in Ukraine⁴, retail prices have remained rather constant. EU member states have adapted measures to shield consumers from rises in wholesale prices—absent these shielding measures, however, the electricity prices would have been expected to be 20% higher since mid-2021⁵. We therefore feel our +/- 20% sensitivity is appropriate. We include the results of the electricity price sensitivity analysis for all modelled user types and European countries in Supplemental Information section B.1.2 Supplementary Table S17. We also include new figures for the electricity price sensitivities as map and chart figures (see Supplementary Figures S9-S11). A further breakdown of the LCOC components can be found in the revised supplementary excel files (see in particular LCOC_Detailed_Results.xlsx). We also discuss newly the implications of the electricity cost sensitivity analysis in the discussion section of the paper (Pg. 15). Similar to the results of the performed sensitivity on TOU tariffs, we observe the biggest changes

⁴ EU Commission, *European Economic Forecast*, Autumn 2021 [https://ec.europa.eu/info/sites/default/files/economy-finance/ecfin_forecast_autumn_2021_ch4_special_issues_2_en.pdf]

⁵ Household Energy Price Index, *Capping Europe's household electricity bill crisis*, Feb 2022 [<https://www.energypriceindex.com/blogs/capping-europes-household-electricity-bill-crisis>]

		to the LCOC in country-specific situations where the grid electricity share is largest. For example, the LCOC shift for residential (grid) sites was greater than the LCOC shift for commercial sites and the LCOC shift for commercial sites was generally greater than the LCOC shift for residential (PV) sites. Finally, we comment shortly on the reviewers question about projected electricity prices over the lifetime of the equipment. As shown in the LCOC equation (1), the electricity cost $C_{electricity}$ does not depend on the charging infrastructure, but is rather a single cost calculated once in a given charging instance. Electricity price projections would therefore not have any effect on the final LCOC value. The LCOC is a snapshot price to charge an electric vehicle at different charging sites that levelizes the infrastructure cost of the lifetime of the equipment, but not the electricity price. Pg. 15 Electricity price sensitivity has become an increasingly important issue in light of rising natural gas prices and the exacerbated energy crisis in Europe as a result of Russia's invasion of Ukraine. Despite the expectation of rising electricity prices, retail prices have remained rather constant since 2019, the year in which this study's price data are based. Our electricity price sensitivity analysis shows that a +/- 20% change in the electricity price can lead to an extreme case +/- 14% change in the LCOC for the "Average user" profile (see Supplemental Information B.1.2).
2.3	Major Aspects 2: I believe there is an issue of comparability when considering households and commercial charging points. While in household charging the final cost for the users should be the LCOC calculated in the paper, in commercial charging the LCOC represents the cost for the supplier, and not for the final users. Thus, these two costs are actually not directly comparable (nor can they be directly summed up for user profiles with a mixed household/commercial charging). Please address this issue.	This is a key distinction that the reviewer brings up with this comment and as well with comment No. 2.7 and No. 2.9. In the following we hope to clarify the justification of the LCOC concept as a final charging cost seen by EV users. Indeed there is a difference between cost for the supplier and cost for the final users. In this study we calculate the LCOC to be the final average cost to the user. At no point do we report costs for the supplier. By taking into account cost of capital as well as sales margins of commercial operators, the LCOC represents the final long-term average cost to the consumer and can therefore be directly summed up for user profiles with mixed household/commercial charging options and is also comparable to average costs of conventional fuels (also brought up in reviewer comment No 2.9). We highlight this point in the discussion section of the manuscript that despite our levelized cost results portraying representative average charging costs in different European countries, station operators may decide to set prices differently, at least in the short term or under imperfect market conditions. Original manuscript text: Here we modelled levelized cost to the end consumer only, but charging infrastructure operators might set prices differently, at least in the short term or under imperfect market environments. Charging prices may stay rather constant at publically available stations in particular in order for station operators to remain competitive. Pricing schemes may additionally depend on how charging infrastructure was historically developed, country or even county-specific subsidies or price regulations, parking fees charged by the station operator, time spent charging (EURO/minute), a flat fee for the entire charging session (EURO/session),

		etc.⁶ These pricing strategies cannot be captured in the LCOC logic, but we do maintain the LCOC to be a representative cost for the final user and not for the supplier. To avoid confusion over the difference between cost for the supplier and cost for the users brought up by the reviewer in comments No. 2.3, 2.7 and 2.9, we have added clarifying statements or comments throughout the manuscript. Specifically in the discussion section we have extended our comments on charging price variance in line with the points made above in this response comment (Pg. 13). Also in the Introduction section we have added lines clarifying that the LCOC represents the final user cost (Pg. 3). Pg. 13 Such pricing schemes may depend on how charging infrastructure was historically developed, national or even sub-national-specific subsidies or price regulations, charging subscription packages, membership status, parking fees charged by the station operator, time spent charging (€/minute) or even a flat fee for the entire charging session.
2.4	Major Aspects 3: In parallel, I believe a comparison of the LCOC calculated for the commercial charging with the average price that is charged to users could provide interesting insights to be included in the discussion.	This is a great point brought up by the reviewer. To integrate this point into the manuscript, we collect additional data points on “true” charging prices posted by major commercial charging station operators in Europe and discuss how they compare to our LCOC results. As a systematic comparison of commercial charging prices on the European level is difficult, we focus rather on a few sources for Germany, UK and Switzerland. We refer the reviewer to the newly added sentences in the discussion section for the specifically collected market prices (Pg. 13). Pg. 13 While a systemic Europe wide overview of current charging prices at publically accessible commercial sites is not readily available, one comprehensive report for Germany in 2020 cites average charging prices from over 300 collected tariffs to be in the range of 0.30-0.35 €/kWh. In the UK in mid-2022, charging prices at publically available stations vary—Ionity stations charge on the high end 0.81 €/kWh while Shell Recharge and Fastned price on the medium to lower end at 0.64 €/kWh and 0.46 €/kWh respectively. Charging prices at Swiss commercial stations are also rather varied and highly dependent on charging location (public parking lot vs. shopping mall vs. on road)—average prices in 2021 fall between 0.23-0.44€/kWh at medium AC powers and 0.29-0.98€/kWh at high DC powers. We find our LCOC results compare aptly to these market values with higher general electricity prices in 2022 likely contributing to the discrepancies for high end prices in the UK and Switzerland. These results confirm our modeled LCOC costs to be in the correct magnitude range with higher general electricity prices in 2022 likely contributing to the price discrepancies between our LCOC results and market values for higher power charging stations in the UK and Switzerland.

⁶ European Parliament, Parliamentary questions, Subject: Price transparency at electric car charging stations, 2019 [https://www.europarl.europa.eu/doceo/document/E-9-2019-003201_EN.html]; EV Roaming Foundation, *Cross-border charging: The necessity of price transparency in EU*, 2020 [https://evroaming.org/overview-price-transparency-in-eu/]; European Alternative Fuels Observatory, *Pricing of Electric Vehicle Recharging in Europe*, 2021 [https://www.avere.org/wp-content/uploads/2021/07/EAF0-Report-Pricing-of-Electric-Vehicle-Recharging-in-Europe.pdf]

		We would like to point out, however, that “true” prices are difficult to ascertain as they can be rather complicated, for example prices can vary much depend on the subscription contract for individual EV users as well as other factors (see response comment No. 2.6). In the LCOC concept we cannot consider these strategic pricing aspects (see also response comment No 2.3). We agree that this is an important discussion point, and a larger comparison of “true” commercial charging prices with our LCOC results would be useful, but is perhaps rather grounds for future research.
2.5	Major Aspects 4: The role of PV generation in households is not clear to me. In Figure 1 the two user profiles “Wallbox” and “Wallbox with PV” seem identical, which makes sense. At the same time, the “Wallbox with PV” will likely not use all its household-electricity from PV directly, but a significant amount of charging time will happen at night, when PV is not available (if not, I expect the shares to be different for the two user profiles). Many EU countries have different rules for self-consumption accounting, so the authors should clarify if they consider actual electricity from PV (and at what price) or an annual balance between production and consumption (without considering different hourly profiles for charging and PV generation).	We thank the reviewer for pointing out an area of potential confusion with our methodology for the “Wallbox with PV” user. Our analysis does indeed account for the fact that for residential EV chargers with solar rooftop PV, a significant amount of charging time happens overnight when PV generation is unavailable. For the Residential PV charging site, rooftop solar PV generation replaces a limited share of the grid electricity during the day, which we base off of an hour-by-hour load profile. The rest of the required charging energy is assumed to come from the grid (i.e. when the sun is not shining). To this end, we do consider produced electricity from solar rooftop PV systems balanced with consumed electricity from the grid according to the weighted shares of PV vs. grid energy charging which are based on hourly load profiles. The PV portion of the cost is based on the average country-specific levelized cost of electricity (LCOE) for solar rooftop PV generation (see Supplementary Table S1). To make this more clear in the manuscript, we have included an additional list of points in the Residential (PV) charging site description in the Introduction section (Pg. 4) as well as some additional clarifying terms in the “Electricity cost” subsection of the Methods section (Pg. 18). We would also refer the reviewer to Supplemental Information section A.4 for a more detailed description of how we handle the electricity cost of residential charging sites with solar rooftop PV. Pg. 4 Residential (PV): home charging, replacing a limited share of the grid electricity during the day with electricity from a rooftop solar PV system, based on realistic hour-by-hour load profiles. Pg. 18 For the residential charging site with on-site PV generation from a solar rooftop PV system in addition to the grid connection, we estimate country-specific hour-by-hour shares of the charging energy from the grid that can be replaced by the PV system during the day and build a weighted cost average.
2.6	Minor Aspects 1: Line 42: it is true that fuel prices are transparent, but so are EV charging prices in most public chargers. Moreover, also traditional fuels show a significant variability (by country, location, highways, brand, etc.), just like for EV charging. This should be maybe further clarified. On the other hand, fuel costs are	We thank the reviewer for bringing up this point. In our response, we would like to perhaps first clarify that our intention was for the distinction between EV and ICEV charging transparency brought up in Line 42 to speak to the inherent variance and thus transparency of cost at different charging sites depending specifically on charging behavior. Charging behavior would dictate whether an EV user prefers a publically available DC fast charging station vs. a private one or even the option to charge at home—with such a variety of charging options, price differences are not always so obvious. Furthermore, for EV users that subscribe to charging packages, the cost to charge at both commercial and residential sites is not immediately clear. In particular, cross border charging price transparency in many EU countries is problematic and has been discussed as an essential hurdle to overcome for

	constantly varying, while in many cases charging costs show a certain level of stability. This aspect may be worth being mentioned as well.	widespread EV adoption⁷. Pricing schemes can also depend on different national regulations or on how the charging infrastructure was historically developed. The situation of ICE car refueling is much less complicated with most variation stemming from differences in national regulation. Consequently, within country variation is rather small. To our understanding, EV charging prices are not very transparent, but in order to avoid confusion, we have added a comment about EV charging contract subscriptions and as well softened the language about EV charging cost transparency in the Introduction section (Pg. 1).
2.7	Minor Aspects 2: Lines 49-51: It is not clear to me if the authors are discussing charging prices for the users or charging costs for the suppliers, as they are often mixing these two concepts. Please double-check these paragraphs to provide a coherent interpretation.	We refer the reviewer to the more detailed response on charging costs for suppliers vs. users in comment No. 2.3 above. We specifically address the issue raised in comment 2.7 by adding clarifying terms and sentences where possible throughout the manuscript as well as a more direct clarification sentence at the end of the fifth paragraph in the introduction (Pg. 1-2).
2.8	Minor Aspects 3: Equation 1: it is not clear to me why the O&M annual costs are discounted while the electricity cost is not. I assume this is coherent with other levelized cost methods in the field. Still, the authors may also want to briefly discuss the effects of this choice.	In the following we give a succinct response to the reviewer, but we do not make additional changes to the manuscript as we feel the logic of our methodology holds: The LCOC gives the cost to charge an EV in any given instant in time. It is a snapshot cost, if you will. The electricity cost is charged to a single EV user for a given charging action. The cost charging infrastructure cost (equipment, installation, O&M), however, is distributed over the lifetime and discounted (a user therefore sees only part of this lifetime cost at each charging action). If we were calculating the total cost of ownership (TCO) of an EV, then it would be appropriate to discount the electricity cost for the total cumulative charging energy over the lifetime of the vehicle, but the LCOC is simply the cost to charge in any given instant. This logic is consistent with paper from Borlaug et al. 2020, upon which we based our methodology.
2.9	Minor Aspects 4: Figure 5: As stated above, LCOC is actually representing the cost for the supplier, not for the final users. Thus, it cannot be directly compared with fuel costs (that include the revenue of the suppliers). Please address this issue.	Again here we refer the reviewer to the more detailed response on charging costs for suppliers vs. users in response comment No. 2.3 above.

⁷ EV Roaming Foundation, *Cross-border charging: The necessity of price transparency in EU*, 2020
[<https://evroaming.org/overview-price-transparency-in-eu/>]